# Cholesterol binding to the sterol-sensing region of Niemann Pick C1 protein confines dynamics of its N-terminal domain

**Vikas Dubey**[1,2], **Behruz Bozorg**[1,2¤], **Daniel Wüstner**[1,3], **Himanshu Khandelia**[1,2,4]*

**1** PhyLife Physical Life Sciences, Department of Physics Chemistry, and Pharmacy, University of Southern Denmark, Campusvej 55, 5230 Odense M, Denmark, **2** Department of Physics Chemistry, and Pharmacy, University of Southern Denmark, Campusvej 55, 5230 Odense M, Denmark, **3** Department of Biochemistry and Molecular Biology, University of Southern Denmark, Campusvej 55, 5230 Odense M, Denmark, **4** MEMPHYS: Center for Biomembrane Physics, Odense M, Denmark

¤ Current address: Department of Astronomy and Theoretical Physics, Lund University, Sölvegatan 14A, 22362, Lund, Sweden
* hkhandel@sdu.dk

**Data Availability Statement:** All input files for reproducing the simulations are available open-access at https://doi.org/10.5281/zenodo.3543512.

## Abstract

Lysosomal accumulation of cholesterol is a hallmark of Niemann Pick type C (NPC) disease caused by mutations primarily in the lysosomal membrane protein NPC1. NPC1 contains a transmembrane sterol-sensing domain (SSD), which is supposed to regulate protein activity upon cholesterol binding, but the mechanisms underlying this process are poorly understood. Using atomistic simulations, we show that in the absence of cholesterol in the SSD, the luminal domains of NPC1 are highly dynamic, resulting in the disengagement of the NTD from the rest of the protein. The disengaged NPC1 adopts a flexed conformation that approaches the lipid bilayer, and could represent a conformational state primed to receive a sterol molecule from the soluble lysosomal cholesterol carrier NPC2. The binding of cholesterol to the SSD of NPC1 allosterically suppresses the conformational dynamics of the luminal domains resulting in an upright NTD conformation. The presence of an additional 20% cholesterol in the membrane has negligible impact on this process. The additional presence of an NTD-bound cholesterol suppresses the flexing of the NTD. We propose that cholesterol acts as an allosteric effector, and the modulation of NTD dynamics by the SSD-bound cholesterol constitutes an allosteric feedback mechanism in NPC1 that controls cholesterol abundance in the lysosomal membrane.

## Author summary

Cholesterol is absorbed from LDL particles in esterified form, and is broken down to free cholesterol in the lysosomes of cells, from where cholesterol must be transported to other cellular compartments such as the plasma membrane. The Niemann Pick type C (NPC) diseases arise from deficient cholesterol transport and result from mutations in the cholesterol transport protein NPC1. Using computer simulations, we show that cholesterol, when bound to one part of NPC1, can control the structural transitions of an 8-nm

**Funding:** The funding sources are as follows: Lundbeckfonden. R82-2011-7280, www.lundbeckfonden.dk to HK; Lundbeckfonden. R82-2011-7280, www.lundbeckfonden.dk to VD; Villumfonden, no grant number, https://veluxfoundations.dk/en to BB; and Danish Council for Independent Research, DFF-7014-00054, https://ufm.dk/en/research-and-innovation/councils-and-commissions/independent-research-fund-Denmark to DW. The funders had no role in study design, data collection and analysis, decision to publish, or preparation of the manuscript.

**Competing interests:** The authors have declared that no competing interests exist.

distant, different part of NPC1 protein called the N-terminal domain (NTD). Such long-range control of protein conformations (allostery), controls a wide range of cellular functions mediated by proteins. Fundamental molecular insights into the function of the NPC1 protein can potentially lead to better pharmaceutical interventions for the NPC diseases.

## Introduction

Cholesterol transport in the bloodstream is mediated by low density lipoprotein (LDL), also called bad cholesterol, as its plasma levels are associated with the risk for developing atherosclerosis [1]. Mammalian cells can internalize cholesterol esters as constituents of LDL by receptor mediated endocytosis of the lipoprotein particles [2]. After internalization, LDL dissociates from its receptor in a sorting endosome and is transferred to endo-lysosomes for degradation. Acid lipase in endo-lysosomes hydrolyzes LDL cholesteryl esters, and the liberated cholesterol is exported from these organelles either to the plasma membrane (PM), the endoplasmic reticulum (ER) or eventually the trans-Golgi network (TGN) [3–7]. The molecular mechanisms underlying this egress of cholesterol from endo-lysosomes are beginning to be understood at the molecular level: Niemann Pick C2 (NPC2) protein, a small endo-lysosomal sterol transfer protein, likely picks up the LDL-derived cholesterol first and transfers it to several proteins in the limiting membrane of endo-lysosomes. ATP-binding cassette transporter A1 (ABCA1), lysosome associated membrane protein 1 (LAMP1) and primarily Niemann Pick C1 (NPC1) protein have been implicated in the next step: the shuttling of cholesterol across the endo-lysosomal membrane for further transport to other compartments [8–11]. NPC1 is a large transmembrane protein whose structure resembles that of bacterial RND membrane permeases and efflux pumps from Gram-negative bacteria [12, 13]. Fibroblasts lacking functional NPC1 or NPC2 hydrolyze LDL-derived cholesteryl esters normally but have a strongly reduced ability to elicit normal regulatory responses [10, 11]. One model proposes that NPC2 binds cholesterol after hydrolysis of LDL CEs in the lumen and transfers it to NPC1 for export from LEs to various target organelles including the ER and the PM [14]. NPC2 binds to the middle loop of NPC1 protruding into the lumen of late endosomes and lysosomes (LE/LYSs), while sterol transfer from NPC2 to NPC1 requires the N-terminal sterol-binding domain (NTD) of NPC1 [15–18]. *In-vitro* experiments directly demonstrated binding of cholesterol and oxysterols to the NTD of NPC1 and transfer of cholesterol between this domain, NPC2 and model membranes [19–21]. Clinical manifestations of NPC disease can be traced back to point mutations in both, NPC2 and NPC1, and a map of mutations has been developed based on the crystal structure of the NTD of NPC1 and of NPC2 and their molecular and functional interactions in sterol transfer [17, 18]. It has been speculated that the transfer of cholesterol from NPC2 to the NTD of NPC1 is a prerequisite for cholesterol insertion into the endo-lysosome membrane, which is covered on its inner site with a glycocalyx preventing free sterol diffusion from NPC2 to the bilayer [22–24]. This, however, would require a large structural rearrangement within the NPC1 core, as its NTD extends about 8 nm from the bilayer [12, 23].

NPC1 contains a sterol sensing domain (SSD), a conserved motif in the transmembrane helices of several proteins being involved in sterol homeostasis [25]. Evidence has been provided that the SSD of NPC1 directly binds cholesterol which is critical for the transport function of NPC1 [26–28], while point mutations in the SSD have been shown to cause NPC1 disease [29, 30]. NPC1 has been recently also identified as the interaction partner for Ebola

virus glycoprotein (EV-GP) [31]. Hydrophobic amines, like U18666A cause an NPC1-like lysosomal cholesterol storage phenotype and prevent Ebola infection, at least partly by binding directly to the SSD of NPC1 [29]. Hedgehog signaling during development of Drosophila depends on an intact SSD of Patched, a close homolog of NPC1, and two binding sites for a close cholesterol homologue, cholesterol hemisuccinate, one in the extracellular domain (resembling the NTD of NPC1) and one in the SSD have been observed in the cryo-EM structure of human Patched [32–34]. Introducing point mutations into the extracellular sterol binding site and the SSD of Patched1 was shown to interfere with sterol binding and to cause large structural rearrangements of the protein compared to the Patched wild-type [34]. While it is well established that sterol binding in the SSD must trigger structural transitions in those proteins, the molecular mechanisms underlying allosteric regulation of SSD containing membrane proteins by cholesterol in the bilayer are not known. Here, we have carried out large-scale molecular dynamics (MD) simulations of NPC1 in the presence and absence of cholesterol in the embedding membrane, addressing in particular the possible propensity of the NTD to tilt towards the membrane surface. Using conformational analysis techniques we show that cholesterol binding to the SSD of NPC1 suppresses conformational changes in the luminal domains that can potentially tilt the NTD towards the lipid bilayer. Statistical analysis based on structural alphabets allowed us to map out the transmission pathway of conformational changes upon cholesterol binding in the SSD. The allosteric transition pathway we discover could control the conformational freedom of the NTD and thereby provide a molecular feedback mechanism by which NPC1 senses cholesterol abundance in the lysosomal membrane.

## Materials and methods

### Construction of the NPC model system for all-atom MD simulations

We used a recent crystal structure (PDB: 5U73) to model NPC [35]. The structure lacks the NTD and TM1. The missing NTD and TM1 were modeled using the cryo-EM structure (PDB:3JD8) [12]. The remaining missing loops were modeled using Modeller [36–39]. During preparation of this manuscript a cryo-EM structure of NPC1 bound to the fungal inhibitor itraconazole was published ([40] (PDB: 6UOX). The RMSD between our model and 6UOX is 1.45 Å. Mannose type N-glycans(GlcNAc$_2$Man$_3$) were added were added using the doGlycans tool [41]. All disulphide bonds described in the literature [12, 35] were retained. Binding of NPC2 to NPC1, sterol transfer between both proteins and NPC2-mediated sterol exchange between model membranes requires an acidic pH as found inside LE/LYSs. [12, 15, 42, 43]. The histidine residues in the luminal region were kept protonated. The first 20 residues were removed from the structure because these residues comprise a signaling sequence for NPC. The protein was embedded in a POPC bilayer containing 500 lipid molecules, which was constructed using the CHARMM-GUI [44–46] and hydrated with ∼70, 000 water molecules. The NPC1 protein was inserted in three types of systems which were conceived: 1) in a bilayer consisting of only POPC, 2) in a bilayer consisting of POPC and cholesterol in the ratio of 80:20% and 3) a bilayer consisting of POPC and with one cholesterol modeled at the alleged binding site in the sterol sensing domain (SSD). These three systems are referred to as POPC, POPC-CHOL and POPC-CHOL-bound. When 20% cholesterol is present, i.e. in the POPC-CHOL systems, a cholesterol molecule binds to the alleged binding site within a few nanoseconds.

### Simulation details and data analysis

All-atom MD simulations were performed using GROMACS version 2016.3 [47–51]. For each of the three types of simulations, 10 replicates with different initial velocities were simulated.

We used the TIP3P water model with Lennard-Jones interactions on hydrogen atoms. A 1.2 *nm* cutoff was used for non-bonded neighbour list, and was updated every 10 steps. The van der Waals interactions were switched off between 1.0 to 1.2 *nm*. Electrostatic interactions were treated with the particle mesh Ewald (PME) method [52, 53]. All systems were minimised with 5,000 steps using steepest decent algorithm, followed by a 10-12 ns equilibration and a subsequent 200 ns production run. The temperature of the system was kept at 310K with the Nosé-Hoover thermostat [54, 55] after an equilibration run which was performed with the Berendsen thermostat [56]. The pressure was kept at 1 bar with the Parrinello-Rahman barostat [57] along with a semi-isotropic pressure coupling scheme. The Linear Constraint Solver (LINCS) [58] algorithm was used to constrain all bonds containing hydrogen. A 1 fs time step was used and trajectories were sampled every 50 ps. Cholesterol-containing simulation systems were unstable with a time step of 2 fs. The data analysis was carried out using GROMACS and homemade scripts. GSAtools [59–61] was used to analyze allosteric interactions and correlated motions in the protein. Snapshots were rendered using Visual Molecular Dynamics (VMD) [62]. Data analysis was performed on a single concatenated trajectory for each of the three systems, unless otherwise mentioned. Trajectories for the center of mass of different domains were extracted by MDAnalysis tools [63, 64] and distance and angular data were calculated by homemade scripts. Covariance and principal components analysis (PCA) was performed using GROMACS.

## Results

### Binding of cholesterol to the SSD

To assess the role played by cholesterol in controlling NPC1's conformational dynamics, we launched three sets of simulations: NPC1 in a pure POPC membrane, in a membrane consisting of POPC with 20% cholesterol and in a POPC bilayer with a single cholesterol bound to the SSD of NPC1. These three systems are henceforth referred to as POPC, POPC-CHOL and POPC-CHOL-bound respectively. Each setup was simulated in 10 replicates for 200 *ns*, giving a total simulation time of 2 *μ*s in each case.

In the POPC-CHOL-bound simulations, cholesterol binds into the hydrophobic pocket created between helices TM3, TM4, and TM5 (Fig 1A). The hydrophobic backbone of cholesterol is surrounded by hydrophobic residues Val621, Tyr628 and Ile635 on TM3, Ile658 and Ile661, Leu665 on TM4 and Ile687 on helix TM5. Glu688 on TM5 forms a hydrogen bond with the cholesterol hydroxyl group. Further molecular features of the cholesterol binding pocket and their significance is further explained in the section *Cholesterol binding to the SSD*. In four out of the 10 POPC-CHOL-bound replicates, the cholesterol molecule escapes from the binding site, and in two of these, it returns to the binding site within 200 ns. However, the cholesterol molecule does not diffuse completely away from the binding site, and remains within 1.7 nm of the SSD (S1 Fig). In the POPC-CHOL simulations, a cholesterol molecule diffuses into the *same* binding site inside the SSD within the first few nanoseconds in each of the 10 simulations, which started with different initial velocity distributions. Such binding is unsurprising, because every fifth lipid surrounding the protein is a cholesterol molecule and the rate constant for binding is bimolecular. We simulated an extra system with a different initial cholesterol distribution in the membrane, and in this case too, cholesterol bound to the SSD on a timescale of 750 ns, although there were several binding and unbinding events to the same binding site before a cholesterol molecule bound stably to the binding site. Accordingly, cholesterol in the membrane will rapidly bind to the SSD of NPC1. Thus, the only difference between the POPC-CHOL and POPC-CHOL-bound systems is the presence of free membrane cholesterol in the former system.

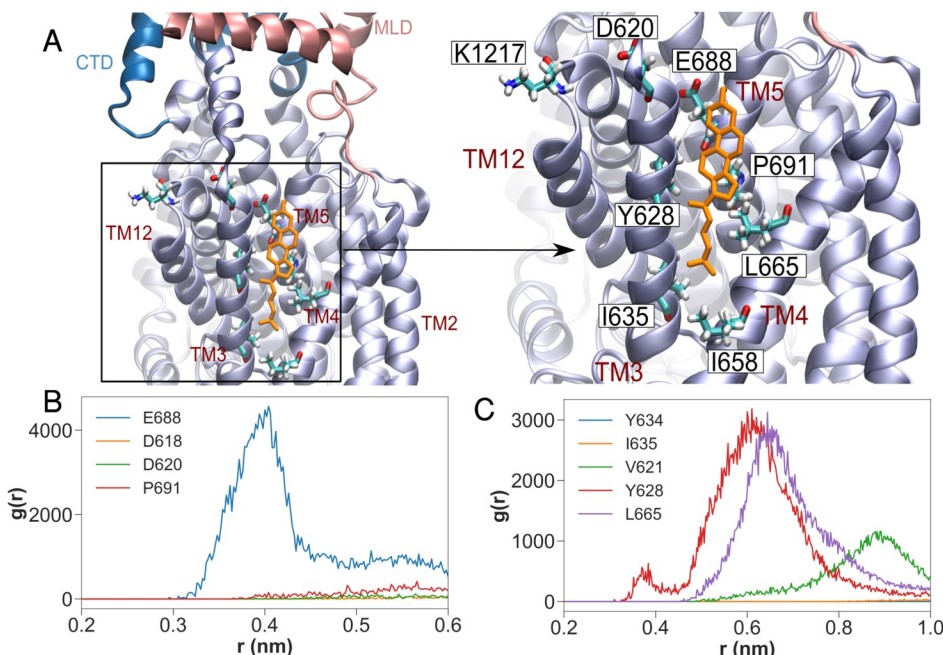

**Fig 1. Binding of cholesterol to the SSD in the POPC-CHOL-bound simulations.** Panel (A) shows the cholesterol hydroxyl group hydrogen bound to Glu688 on top of TM5. (B) Radial distribution functions (RDFs: normalised histograms of distances between atom selections, averaged over time) between the hydroxyl group of cholesterol and the side chains of nearby residues over all 10 concatenated POPC-CHOL-bound trajectories. (C) RDFs between the center of mass of cholesterol and hydrophobic side chains in TM3 and TM4.

## Cholesterol-dependent dynamics of the entire NPC1 protein

The root mean square deviation (RMSD) measures the overall deviation of the protein backbone from the starting structure, and thereby informs about the overall dynamics of the protein. For the POPC system, higher RMSD values compared to the other systems indicate pronounced flexibility of the entire protein. Only the RMSD distribution of POPC has a long tail between 1.0 and 1.4 nm (see S2D Fig), indicating significant deviation from the starting structure. Visual inspection (S1 Video) shows that the NTD flexes down drastically (towards the membrane) in 3 out of 10 POPC simulations (see snapshots in Fig 2). Once bent, the NTD does not swivel back to the upright state on our simulation timescales. We use geometric and conformational analysis to quantify the visually observable flexing of the NTD.

## Dynamics of the N-terminal domain

Fig 3 shows the distance between NTD and upper leaflet of the membrane. Both the distance between the center of masses (Fig 3A) and the minimum distance (Fig 3B) between the NTD and the upper leaflet of the bilayer are measured. According to both measurements, the NTD is closer to the membrane in the absence of cholesterol. The membrane becomes thicker by 0.4 nm when it contains 20% cholesterol (Fig 3C) and this can affect the distance between protein NTD and membrane upper leaflet. Assuming that the upper leaflet is elevated by half of the thickness increase with 20% membrane cholesterol, we expect a 0.2 *nm* relative decrease in the average distances for POPC-CHOL in Fig 3A. On the contrary, the distance distribution for POPC-CHOL is shifted to higher values by about 1 *nm* compared to POPC in the distance distribution (Fig 3A). The distance distribution for the POPC-CHOL-bound simulations demonstrates the importance of binding of only one cholesterol molecule in the binding site of

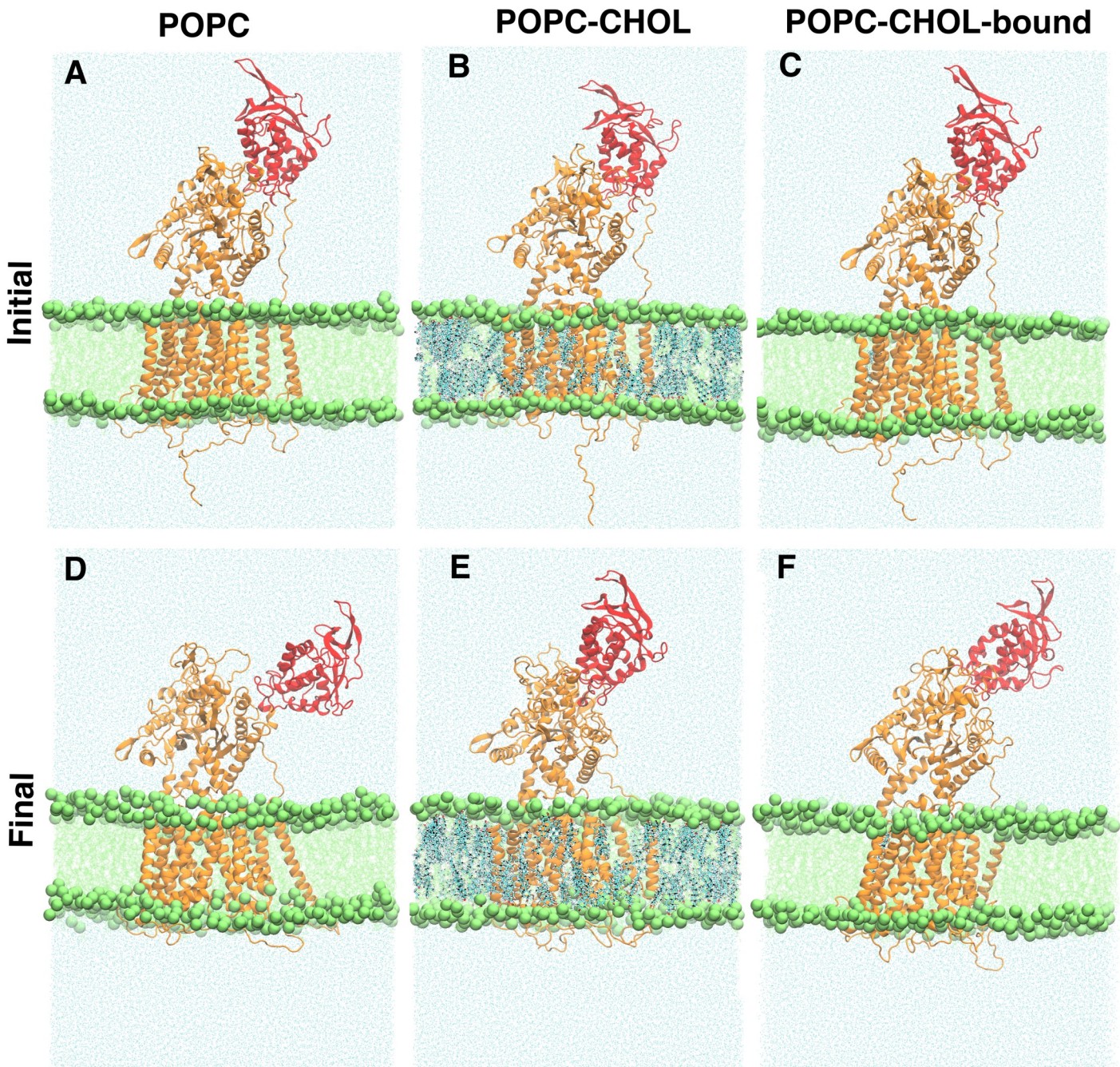

**Fig 2. Representative initial and final simulation snapshots for one of the POPC (A and D), POPC-CHOL (B and E), and POPC-CHOL-bound (C and F) simulations.** The NTD is depicted in red.

NPC1. In this case there is no additional cholesterol in the membrane, and thus the membrane properties and thickness are the same as 20% membrane cholesterol. Yet, the distance between the NTD and the membrane is significantly higher for POPC-CHOL compared to POPC.

To quantify the flexing of the NTD towards the membrane, we measure the angle between two vectors: One vector connecting the center of mass of the middle luminal domain (MLD) with that of the NTD (vector 1) and another vector connecting the center of mass of the

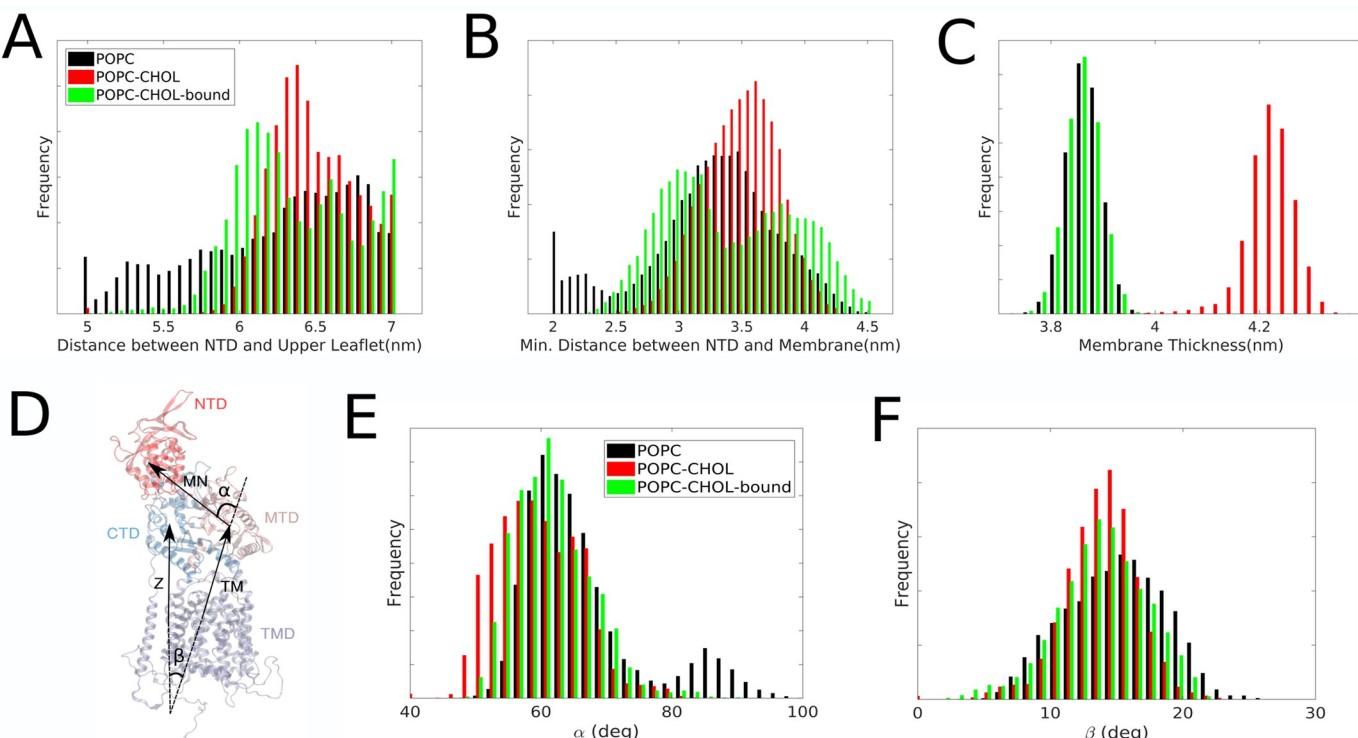

**Fig 3.** A) Distribution of the distance between the center of mass of the NTD and the center of mass of upper leaflet of the membrane. B) Distribution of the minimum possible distance between membrane surface and the NTD C) Distribution of the thickness of the membrane estimated by measuring the distance between the center of masses of the phosphate groups of the upper and lower leaflets of the membrane. Analysis was performed on a concatenated trajectory of 10 independent 200 ns simulations. D) $\alpha$ is calculated as the angle between the vector connecting the center of mass of the transmembrane domain (TMD) and the middle luminal domain (MLD) and the vector connecting the center of mass of the MLD to the center of mass of the NTD. $\beta$ is defined as the angle that the vector connecting the center of mass of TMD and MLD, makes with the bilayer normal ($z$ axis). E) and F) The distribution of the angles $\alpha$ and $\beta$ respectively. The differences in the $\alpha$ distributions are statistically significant (S3 Fig).

transmembrane domain (TMD) with that of the MLD (vector 2; Fig 3D). The distribution of this angle, designated $\alpha$ is centered at about 60˚ for all three systems, whereas $\alpha$ values are slightly lower for the POPC-CHOL system indicating a more upright orientation of the NTD in this system (Fig 3B). For the POPC system, a second population of $\alpha$ centered at 90˚ is found, indicating that the NTD approaches the membrane when no cholesterol resides in the bilayer. To rule out overall rigid-body protein tilting which can also bring the NTD towards the membrane, we analysed the distribution of the angle between vector 2 and the bilayer normal (Fig 3D). There are minor differences in the three distributions. The angle distribution corresponding with the second case (POPC-CHOL) is slightly sharper which is related to the membrane becoming stiffer in the presence of cholesterol, thus reducing the overall mobility for the NPC1 protein Fig 3C. The distribution for POPC is shifted slightly to higher angle values, with a peak at about 16˚ compared to about 14˚ for POPC-CHOL-bound, a minor difference related to the overall increased flexibility of the protein in the POPC system, as described in the following sections. The angle analysis in Fig 3D and 3E are consistent with the distance analysis in Fig 3B and 3C, demonstrating that the NTD is more flexible and bends towards the membrane interface in the absence of cholesterol in the bilayer. Surprisingly, the presence of a single cholesterol bound to the SSD suppresses the NTD movement nearly as much as 20% cholesterol in the membrane. To further quantify the conformational flexibility of the NTD, we carry out a principal component analysis (PCA) to extract global collective motions.

## Conformational dynamics via principal component analysis (PCA)

**Principal components analysis.** After removing center of mass translation and rotation, the covariance analysis was performed for the protein backbone atoms. First, a covariance matrix $C$ is extracted from atomic fluctuations

$$C = \langle (\mathbf{x}(t) - \langle \mathbf{x} \rangle)(\mathbf{x}(t) - \langle \mathbf{x} \rangle)^T \rangle \tag{1}$$

where $\mathbf{x}$ is a $3N$-dimensional column vector describing the coordinates of the $N$ protein backbone atoms, and $\mathbf{x}(t)$ are the positions at time $t$. The triangular brackets indicate an ensemble average. $C$ is a $3N$ x $3N$ symmetric matrix, and is then diagonalised by a coordinate tranformation $R$

$$C = R \Lambda R^T \tag{2}$$

$$\Lambda = R^T C R \tag{3}$$

This orthogonal transformation transforms $C$ into a diagonal matrix $\Lambda = diag(\lambda_1, \lambda_2, \ldots \lambda_{3N})$, which contains the eigenvalues $\lambda_i$ of the covariance matrix $C$. The $i$th column of $R$ contains the $i$th eigenvector $\mathbf{r}_i$ with the corresponding eigenvalue $\lambda_i$.

The global collective motions of the trajectory can then be obtained by projecting the trajectory ensemble onto individual eigenvectors, to obtain the principal components $\mathbf{p}_i$, $i = 1, 2, \ldots N$ by taking an inner product between the transpose of the eigenvector and the atomic fluctuation.

$$p_i(t) = \mathbf{r}_i \cdot (\mathbf{x}(t) - \langle \mathbf{x} \rangle) \tag{4}$$

Each trajectory snapshot, thus projected on two different eigenvectors, yields one point on the point cloud distributions shown in Fig 4.

The covariance analysis reveals that the first two eigenvalues account for 55%, 43% and 48% of the fluctuations (see Fig 4E) in the POPC, POPC-CHOL and POPC-CHOL-bound cases, respectively. Motions along these degrees of freedom constitute the largest amplitude collective modes determined from the MD trajectories. The amplitude (square root of the eigenvalue) of collective motions is much larger for the POPC simulations compared to the cholesterol-free simulations (Fig 4). Due to their dominance, we further analyse the global collective motions along the first two eigenvectors. Each protein conformation is projected upon the two dimensional space formed by the first two eigenvectors to obtain a 2D-scatter plot of projected conformations (Fig 4). Dense regions in this plot represent key protein conformations sampled by global collective motions, likely to be free energy minima in the protein conformational landscape. The projection map thus serves as a visualization of the free energy landscape of protein conformations.

Near each axes, we show conformations of the protein which are obtained by interpolation between the extremes of the POPC trajectory when projected upon its first two eigenvectors. These extremes represent the maximum amplitude motions along a specific global collective motion, please refer to S2 Video. On the opposing axes, a histogram for the point cloud distribution is shown. For the POPC trajectories, the most dominant global collective motions both lead to a conformational state where the N-terminal flexes and approaches the membrane. These are conformations which are closer to the origin in Fig 4A and 4C. The cluster of conformations marked (1) in Fig 4A represents the flexed state and is present in the POPC system, but not in the other two systems. In Fig 4B the projection of first two eigenvectors of three types of simulations on the same eigenspace as in A is shown. The similarity between the two cases with cholesterol is apparent. As a consistency check we repeat the same analysis, but now

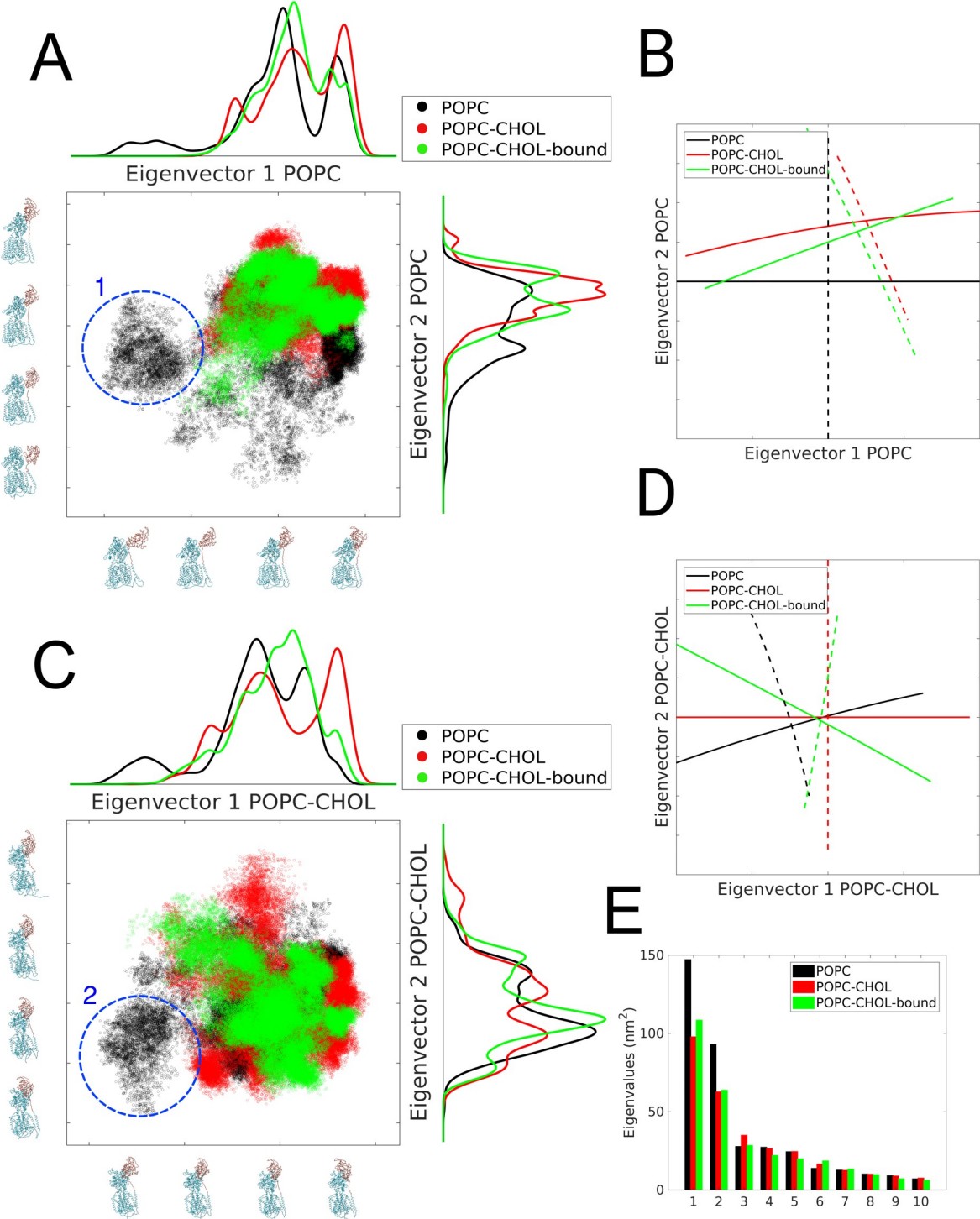

**Fig 4. Principal component analysis.** A) Scatter plot of the all trajectory frames projected on first two eigenvectors of the POPC simulation. Near each axes, we show conformations of the protein which are obtained by interpolation between the extremes of the POPC trajectory when projected upon its first two eigenvectors, i.e. along the dominant global collective motions, please see (S2 Video) B) The solid and dashed lines represent the first two orthogonal eigenvectors, projected upon the eigen-space of the POPC systems. C and D) Similar to A and B respectively, except the eigenspace is formed by the first two eigenvectors of the POPC-CHOL simulations. The cluster of conformations marked 1 in (A) and 2 in (C) represent the flexed state, and are only observed in the POPC systems. The analysis is performed on concatenated trajectories of 10 independent simulations in each case. (E) The first 10 eigenvalues obtained in the PCA are plotted for the three systems.

projecting all trajectories onto the eigenspace formed by the first two eigenvectors of the POPC-CHOL simulations (Fig 4C). Global collective motions along the first eigenvector bring the NTD closer to the membrane, and the cluster of conformations marked (2) in Fig 4C which represents the flexed state, is again present only in the POPC system.

In summary, the PCA analysis shows that NPC1 explores a larger part of the conformational space when embedded in a membrane without cholesterol, in particular with respect to the movement of the NTD towards the membrane. When a single cholesterol molecule is bound to the SSD, the molecular motion of the entire protein becomes more confined, and this effect is further pronounced upon adding 20% cholesterol to the bilayer. The fact that a single cholesterol molecule in the SSD already confines the global collective motions substantially, suggests that an allosteric interaction between the SSD and the luminal domains of NPC1 must be responsible for the difference in the properties of POPC and cholesterol-containing conformations.

## Allosteric links between the NTD and the SSD

How does cholesterol-binding to the SSD alter the conformational dynamics of the NTD? To understand the signal transmission mechanism between the allosteric site in the SSD and other domains, in particular the NTD, we use GSAtools [59–61] a software based on a structural alphabet of protein secondary structures. Four consecutive amino acid residues are combined into a fragment ($f$) such that two consecutive fragments share three residues. Each fragment is uniquely described by a set of three independent angles which can change during the course of an MD simulation. A structural alphabet is used to describe prototypical backbone conformations based on these three angles [60]. During an MD simulation, each fragment gets assigned to a structural alphabet based on the measurement of the three angles at a specific time step. Thereafter, each protein conformation sampled during an MD trajectory can be described as a sequence of structural alphabets, akin to the description of a protein sequence by amino acid alphabets. A network of correlated motions can then be modelled as transitions between these protein fragment descriptors. Finally, an allosteric signal propagation is quantified by the extent of information exchange through such a network. The method has been successfully used to investigate allosteric signalling in proteins such as cardiac myosin [65] and the Ras GTPase [66].

Network and important fragments are identified via two quantities $I^n_{LL}$ and $Ml_n$ [60]. Both of the quanties are different flavours of mutual information. Mutual information quantifies the mutual dependence between two variables. In other words, it measures the information that can be obtained about other variable by observing changes in one variable. In case of GSAtools, normalised mutual information between two fragments $i$ and $j$ ($I^n_{LL}$) is obtained by:

$$I^n_{LL}(C_i; C_j) = \frac{I(C_i; C_j) - \epsilon(C_i; C_j)}{H(C_i, C_j)} \qquad (5)$$

$C_i$ and $C_j$ are columns $i$ and $j$ in the string alignment. $I(C_i;C_j)$ is the mutual information and $H(C_i, C_j)$ is the joint entropy of two fragments. $\epsilon(C_i;C_j)$ represents the error term arising from finite size error. $I^n_{LL}$ measures the local correlation motions of fragments. Global structural changes are extracted by projecting the simulation trajectory onto the eigenvector obtained through principal components analysis and then, correlation between local and global motions is measured by $MI^n$:

$$MI^n(C_i; sPC_j) = \frac{I(C_i; sPC_j)}{H(C_i, sPC_j)} \qquad (6)$$

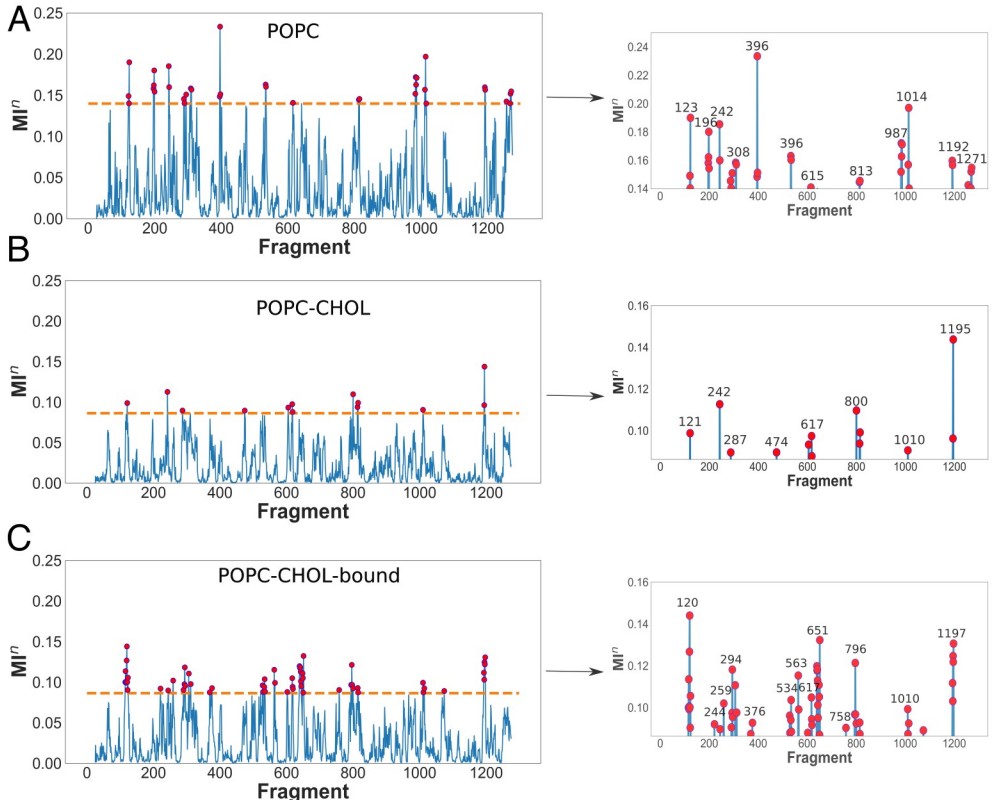

**Fig 5. GSAtools analysis.** The left panels show the distribution of the mutual information $MI_n$ obtained by correlation of local and global motions (first eigenvector) for all protein fragments. Rows A, B and C show data for POPC, POPC-CHOL, POPC-CHOL-bound simulations respectively. The fragments with the top 40% $MI_n$ values are shown on the right panels. In the case of the POPC simulations, the three replicates analysed are those where significant NTD flexing towards the membrane is observed. In the other cases, concatenated trajectories from 10 independent simulation replicates are used. The dashed line in A, B and C indicates the 40% cutoff. Note that the scale of the ordinate (y-axis) is different on the right panels.

Where $sPC_j$ stands for the global states associated with the respective $j$th eigenvector. As before $I(C_i; sPC_j)$ is the mutual between $C_i$ and $sPC_j$ and $H(C_i, sPC_j)$ is their joint entropy. For more details please refer to [60].

We first calculate the mutual information $MI_n$ obtained by correlating the projection of the first eigenvector (Fig 4) to the local motions of individual fragments using GSAtools [60, 61]. Fragments with high $MI_n$ values contribute significantly to global collective motions. To filter out less-important fragments (with low $MI_n$), we extract fragments within the top 40% $MI_n$ values. The 40% cutoff is chosen to relate the $MI_n$ values to the allosteric network described below. We note that the $MI_n$ values are significantly higher in POPC compared to the POPC-CHOL and POPC-CHOL-bound systems (Fig 5). The residues corresponding to each fragment are shown in Table 1.

To extract an allosteric path from the SSD to the NTD, we select nodes (fragments) and edges (connections between fragments) which have an information $I_{LL}^n$ weight of more than 50%. $I_{LL}^n$ estimates the extent of correlation between the conformations of pairs of protein fragments. A graph is generated from the remaining nodes and edges, after implementing the 50% cutoff. The nodes *f242* and *f618* are fed as the endpoints to this network, Thereafter, a shortest path, deemed as the allosteric path, is retrieved between these two fragments by choosing a

**Table 1. The residues associated with specific fragments of allosteric path obtained by GSAtools.**

| Fragment | Residues |
|----------|----------|
| f242 | Asp242, Cys243, Ser244, Ile245 |
| f396 | Asp396, Gln397, His398, Phe399 |
| f534 | Leu534, Gly535, Thr536, Phe537 |
| f618 | Asp618, Ser619, Asp620, Val621 |
| f1014 | Gly1014, Gly1015, His1016, Ala1017 |

distance-based cutoff, i.e. fragments that are too far apart in Cartesian space must have an intermediate fragment to be part of the same allosteric pathway.

The most probably allosteric path between the SSD (fragment *f618*, residues 618-621) and the NTD (fragments *f241-f242*, residues 241-246) passes through fragment *f534* in the MLD, *f396* in the MLD and *f1014* in the CTD (Fig 6). In the POPC-CHOL and POPC-CHOL-bound simulations, these regions are either not connected to the network, or are connected with an insignificantly weighted connection. To emphasize the features that we extract using GSA-Tools, we only analyse the three POPC simulations where the N-terminus flexes significantly towards the bilayer surface.

The local motions of fragments or amino acid tetrads which are part of allosteric signalling pathways must also have a high correlation to the global collective motion which flexes the NTD towards the membrane. Thus, the $MI_n$ values of such fragments should be high to reflect significant allosteric connections. Several fragments which are present in the allosteric path have high $MI_n$ values in the POPC simulations. Fragments *f396*, *f534* and *f1014* are exclusive to POPC and also appear in the allosteric path. Most notably, fragment *f396* has the highest $MI_n$ scores amongst all analysed trajectories. Thus, the amino-acid residues in the range 396-399 are likely to be pivotal in the flexing of the N-terminus. Compared to other fragments in the allosteric path, the fragment *f618* in the cholesterol binding site has a relatively lower $MI_n$ score of 0.126 in POPC. Importantly, the score is higher compared to the POPC-CHOL and POPC-CHOL-bound cases.

## Clinical mutations along the allosteric path

If specific amino acid residues or regions around these residues are important for the functional dynamics of the protein domains, it is also likely that mutations in these amino acids lead to compromised cholesterol transport and disease. More than 100 disease-causing mutations have been mapped onto the different domains of NPC1 [35]. Several of these mutations are close to the allosteric network that we describe. For example, several mutations leading to a clinical phenotype of NPC disease and strongly impaired cholesterol esterification in a cellular functional assay have been described for two residues close to *f396* in the MLD between TM2 and TM3; namely P401L, P401T, R404Q, R404P and R404W [67–69]. Although residues 401 and 404 do not strictly lie within fragments *396* (residues 396 to 399), mutations in 401 and 404 are highly likely to influence the conformations of the nodes in the allosteric path, and, thus, affect the dynamics of the NTD. In fact, the R404Q mutation is among the most frequent mutations in 143 patients studied by Park et al. (2003) [69]. Also in the MLD are two mutations, which either belong to *f534* in our analysis (F537L) or are very close (P543L), both being associated with a severe infantile clinical phenotype of NPC disease [70]. At the interface of SSD and MLD, mutations in or close to *f618* have been found to lead to severe manifestations of NPC disease (i.e., R615C and R615L as well as E612D) [69–71]. In the CTD, three mutations belonging to *f1014* have been described (i.e., G1012D, G1015V and H1016R), which result in

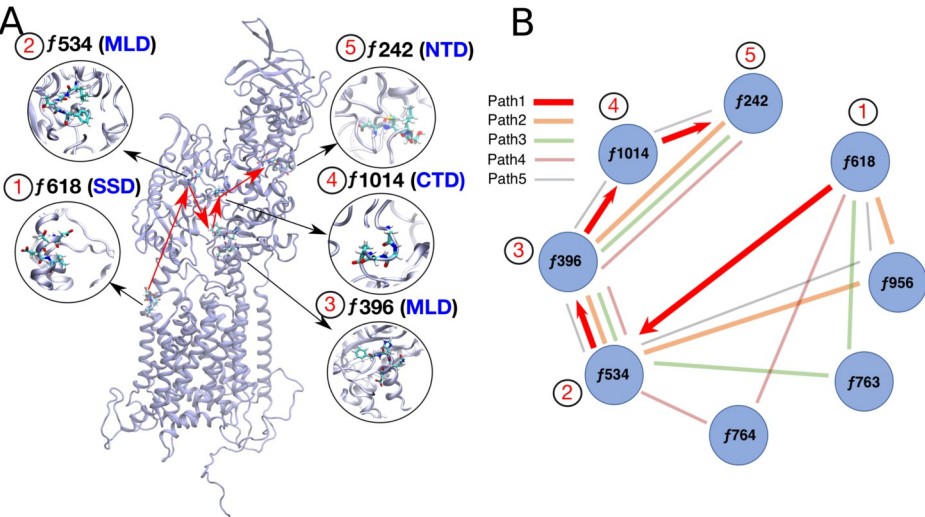

**Fig 6. GSAtools analysis of allosteric networks for the 10 POPC simulations.** (A) The key nodes through which the allosteric signal passes are connected by red lines, and marked by numbers 1 through 5 sequentially. The top 5 shortest paths are shown in the graph in panel (B). The shortest path is shown by thick red arrows.

strong lysosomal accumulation of non-esterified cholesterol in patient fibroblasts and also lead to juvenile NPC disease with psychomotoric retardation and early death [69, 72, 73]. Similarly, in the NTD, severe missense mutations are found in or close to *f242* (i.e. D242H, D242N and P237S) [67, 68, 74]. Importantly, that region of the NTD is not involved in sterol binding but located on the surface of this domain [67, 75]. Together, these results show that the allosteric signaling path that we predict for NPC1 can be mapped to known clinical mutations of the protein, thereby highlighting the relevance of our analysis.

The allosteric network provides the initial clues for further investigation of residue-residue interactions which link cholesterol binding to the suppression of the conformational flexibility of the NTD. To obtain further insights into the specific residue-residue interactions responsible for the signal transmission across the network, we first analyse the conformational changes that occur in the cholesterol binding pocket.

## Cholesterol binding to the SSD

As described earlier, cholesterol binds to the SSD between helices TM3, TM4 and TM5 and acts as a hydrophobic glue, which alters the orientation and dynamics of several side chains in the TMD. The radial distribution function between Glu688 and the cholesterol hydroxyl group indicates that a hydrogen bond between these two moieties (Fig 1B) anchors the cholesterol headgroup to the SSD. Glu688 is either conserved, or is replaced by a Glutamine (S4 Fig), which is also capable of making strong hydrogen bonds with cholesterol. Note that a Glutamine residue resembles a protonated version of Glutamate, and the side chain of Glutamine can still form a hydrogen bond with the hydroxyl group of cholesterol.

Glu688 lies close to Pro691, which is a highly conserved residue in NPC1, SCAP, Patched, and HMG-CoA reductase. The Pro691Ser mutation abolishes the cholesterol transfer function of NPC1 [76, 77]. Proline is a helix breaking amino acid, and the Pro691Ser mutation is likely to significantly alter the conformation of helix TM5, such that the binding of cholesterol to Glu688 becomes unfavourable. The Tyr634Ser and Tyr634Cys mutations also have a compromised cholesterol transfer function [77]. Tyr634 lies at the cytoplasmic end of of helix TM3,

and although distant from cholesterol, is involved in a $\pi$-$\pi$ stacking with residue Phe1207 on helix TM12 (S9 Fig). The Tyr634 sidechain is anchored to TM12 by means of this stacking, and by a hydrogen bond between the Tyr634 hydroxyl group and the backbone of Leu1204 (S9 Fig). The stacking is stronger in the presence of cholesterol. The disruption of such stacking by removing the aromatic ring in the mutations Tyr634Ser and Tyr634Cys can alter the conformation of TM3, and thereby compromise the binding of cholesterol to the SSD. Binding of cholesterol triggers a network of conformation stabilizing interactions, starting from a salt bridge between Asp620 on TM3 and Lys1217 on TM12. This network is analysed in the following section.

## Impact of SSD-bound cholesterol on residue-residue interactions

For each set of simulations (POPC, POPC-CHOL or POPC-CHOL-bound), we construct a distance matrix between all residue pairs, averaged over all 10 trajectories (or in some cases, 3 trajectories for POPC, as described below). For illustration purposes, S6 Fig shows the distance matrix for the entire protein for the POPC simulations. S7 Fig shows the distance matrix of the POPC simulations, subtracted from the distance matrix of the POPC-CHOL simulations. The distance matrices shown in Fig 7A, and later in Figs 8A and 9A are extracted from S7 Fig. Note that this matrix analysis only provides differences in average distance over the entire trajectories. The differences in interactions between pairs of residues must be further corroborated

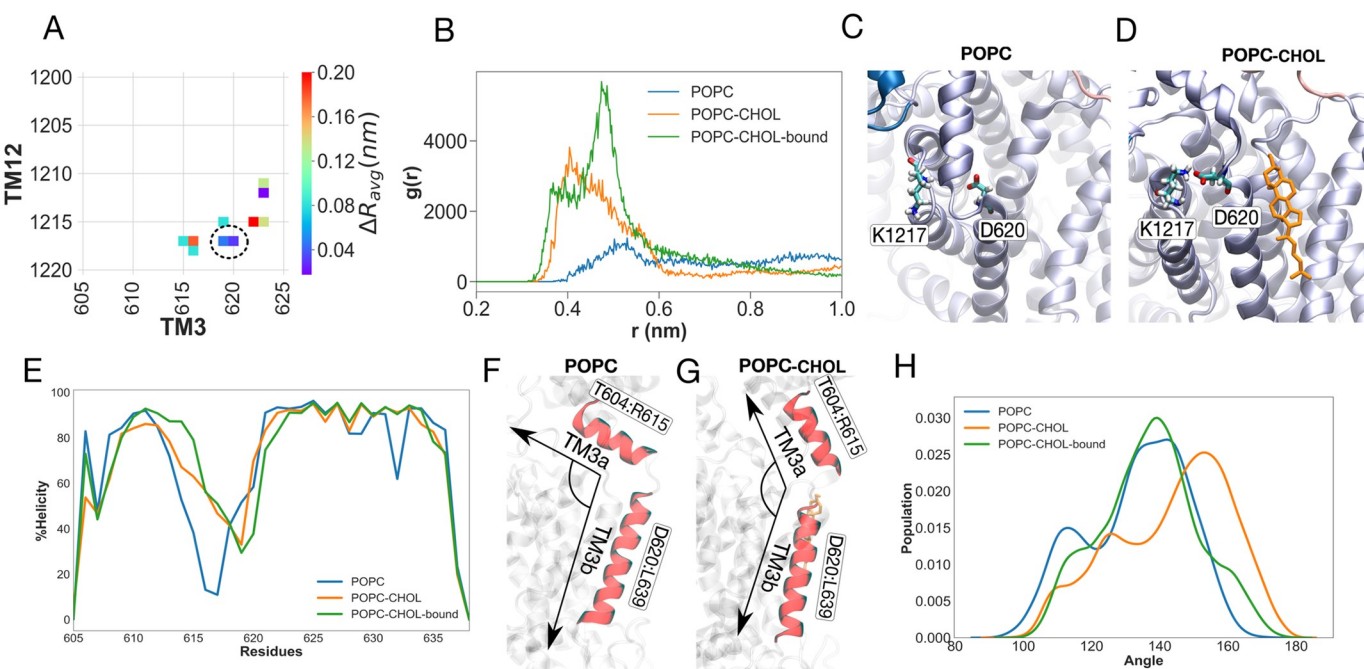

**Fig 7. The binding of cholesterol to the SSD induces changes in the Asp620-Lys1217 salt bridge.** (A) The average distance difference matrix between TM3 and TM12. The difference is obtained from subtracting average distances between residue pairs in the POPC-CHOL simulations from the average distances in the POPC simulations. The hotspots are all in TM3 region between residues 615 and 623. (B) Radial distribution functions reveal a key salt bridge between Asp620 and Lys1217, which is weaker in the POPC simulations. The interaction is annotated by a black dotted circle in (A). (C) and (D) Simulations snapshots from the POPC and POPC-CHOL simulations, respectively. (E) Percentage helical content for TM3 compared for all three sets of simulations. Depiction of how the angle between TM3a (residues Asp620 to Leu639 and TM3b (residues Thr604 to Arg615) differs between the POPC (F) and POPC-CHOL (G). (H) The distribution of the angle shown in (F) and (G). Higher values of the angle (closer to 180 degrees), represent an unkinked, continuous helix. The lower the value of the angle, the greater the kink between the two helices. The POPC distribution is shifted towards the left, denoting increased kinking of TM3. Note that for the distribution, 10 POPC-CHOL and POPC-CHOL-bound simulations are compared to the 3 simulations where the NTD flexing is most apparent. Please see S8 Fig for the comparison between all 10 simulations.

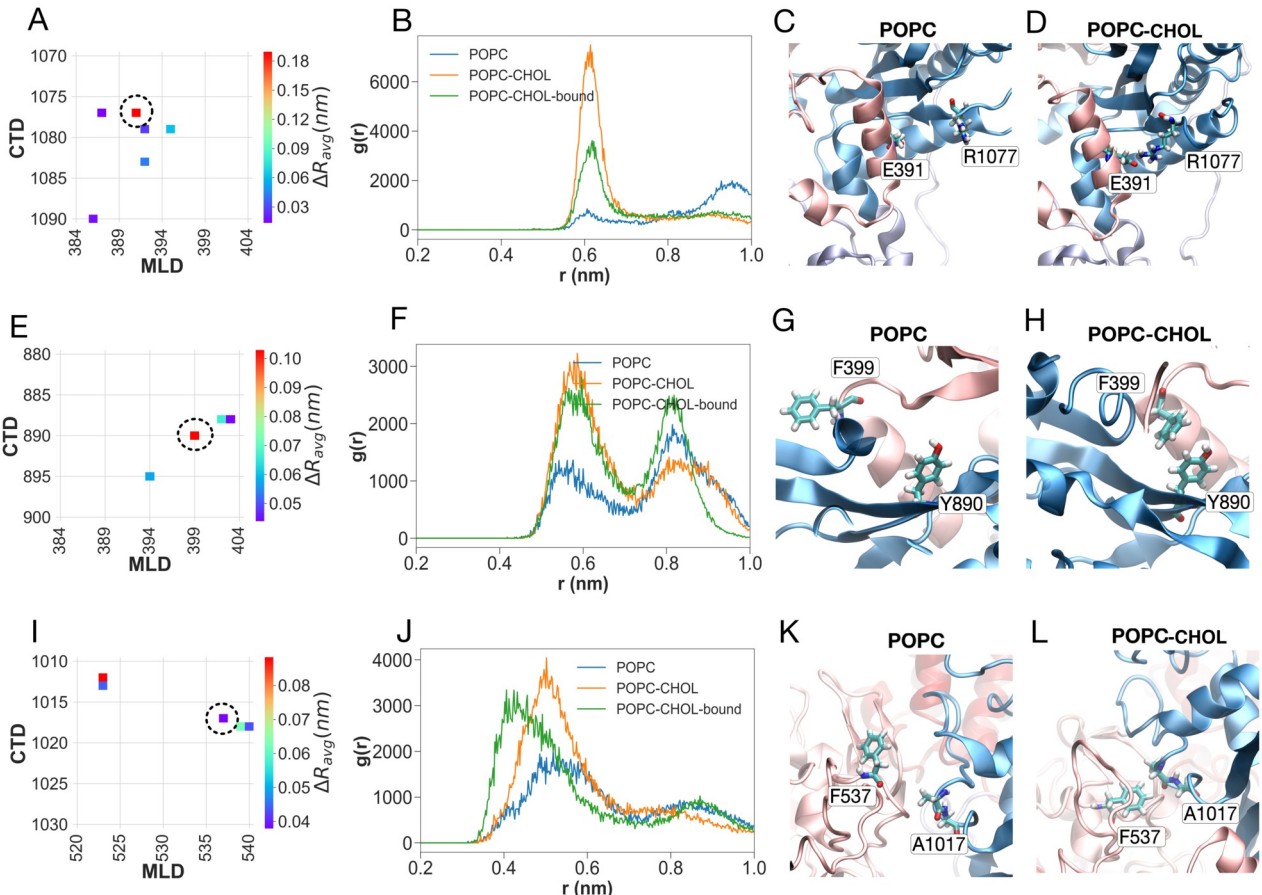

**Fig 8. Interactions between the MLD and CTD are weaker in the POPC simulations.** Panels (A), (E) and (I) show the average distance difference matrices between the specific regions of interest in the MLD and CTD. Panels (B), (F) and (J) in the second column show the radial distribution functions between the specific pairs of residues. In each case, interactions are weaker for the POPC simulations. Panels (C), (G) and (K) in the third column show simulations snapshots from the POPC simulations. Panels (D), (H) and (L) in the fourth column show simulations snapshots from the POPC-CHOL simulations. Residues used in the calculation of radial distribution functions are highlighted.

using radial distribution functions. In the following discussion, we focus on specific contacts that are significantly altered upon binding of cholesterol in the POPC-CHOL and POPC-CHOL-bound simulations. In the presence of cholesterol (POPC-CHOL and POPC-CHOL-bound simulations), Asp620 on helix TM3 and Lys1217 on TM12 form a salt bridge (Fig 7B, 7C and 7D), which is weaker in the absence of cholesterol (POPC simulations). Asp620 is part of fragment *f618* in the allosteric path. The salt bridge stabilises the long helix TM3, which connects the C-terminal end of the MLD to the transmembrane helix bundle. In the absence of cholesterol, the salt bridge is significantly weaker and this correlates with the unfolding of TM3 near residues 615 to 618 (Fig 7E). TM3 splits into two helices upon kinking: the TM helix between residues 620 and 638 (TM3a), and a linker helix between residues 606 and 615 (TM3b) (Fig 7F). When the NTD is flexed, TM3b is sometimes almost parallel to the membrane surface (Fig 7H), thus reducing the overall linker length between the MLD and TMD, earlier mediated by the full helix TM3 (residues 606 to 638). Importantly, the difference in the kinking propensity of TM3 only becomes apparent when all 10 replicates of the POPC-CHOL and POPC-CHOL-bound simulations are compared to only the three POPC simulations where flexing of the NTD is most apparent (S8 Fig). In the POPC-CHOL simulations, the

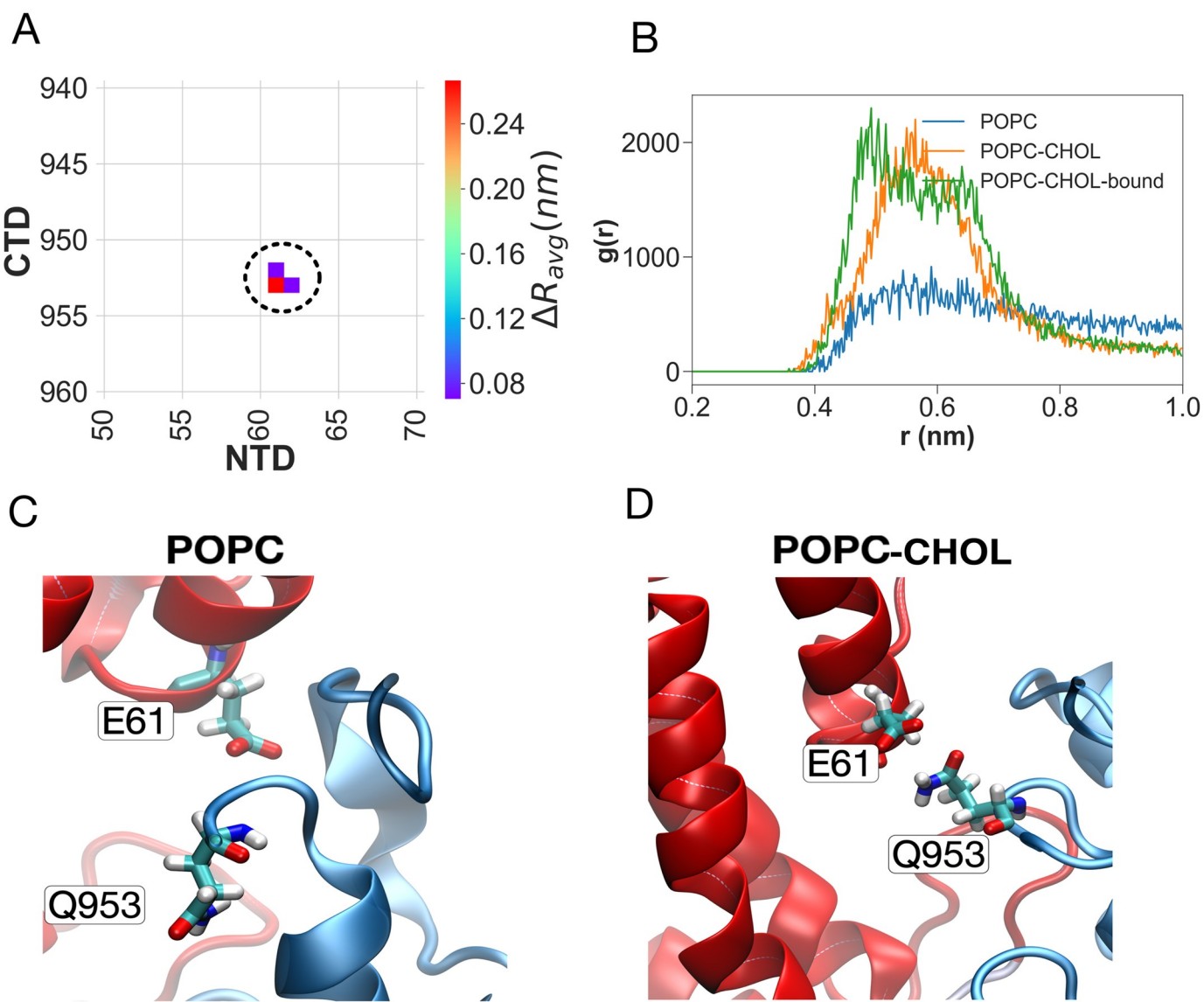

**Fig 9. Interactions between the NTD and CTD are weaker in the POPC simulations.** Panels (A) shows the average distance difference matrix for the specific region of interest. Panel B shows the radial distribution function between residues Glu61 and Gln953. Panels C and D show simulation snapshots from the POPC and POPC-CHOL systems, respectively.

membrane is thicker, and this likely contributes to the reduced kinking propensity of TM3, compared to that in the POPC-CHOL-bound simulations. Therefore, the angle distribution for the POPC-CHOL simulations in Fig 7F is shifted to the right. Note however, that the kink is still significantly higher for POPC, compared to the POPC-CHOL-bound simulations, indicating that free cholesterol content in the membrane alone does not drive changes in the conformation of TM3. The kinking of TM3 results in a rigid body movement of the MLD, which eventually leads to the flexing of the NTD, by the mechanism further described below.

The greater movement of the MLD in the absence of cholesterol weakens several key contacts between the CTD and the MLD which are otherwise present when a cholesterol molecule is bound to the SSD. Two of these contacts lie on the MLD helix between residues 387 and 399. The first of these is the salt bridge between Glu391 on the MLD and Arg1077 on the CTD

(Fig 8A through Fig 8D). The charge reversing E391K mutation has been recently identified to cause NPC disease, and is certain to break the salt bridge between Glu391 and Arg1077 [78]. The second contact is π-π stacking between residue Phe399 on the MLD and residue Tyr890 on the CTD (Fig 8E through Fig 8H). Glu391 is close to, while Phe399 lies exactly on fragment *f396* implicated in the allosteric network. The third contact is a hydrophobic network of inter-actions between residues Phe537 on the MLD and residues Ala1017 and Ala1018 on the CTD (Fig 8I through Fig 8L). The hydrophobic interactions are strengthened in the presence of an SSD-bound cholesterol. The loss of the interactions between these two regions in the absence of cholesterol is manifest in the appearance of both fragment *f534* and fragment *f1014* in the allosteric pathway in the POPC system.

Disruption of the CTD-MLD contacts in the absence of an SSD-bound cholesterol in turn severs contacts between the NTD and CTD. A key interaction which breaks concomitant with the flexing of the NTD is a hydrogen bond between Glu61 on the NTD and Gln953 on the CTD, which keeps the NTD anchored to the CTD in the simulations with cholesterol (POPC-CHOL and POPC-CHOL-bound), but not in the simulations without cholesterol (POPC) (Fig 9). Cys956, which is close to Gln953, is disulfide-bridged to Cys1011, which is close to the frag-ment *f1014* implicated in the allosteric path. Disruption of the hydrogen bond between Glu61 and Gln953 is likely to be important in the flexing of the NTD, and increases the overall dis-tance between the NTD and the CTD (S10 Fig).

A 20-amino acid proline-rich linker starts from residue 243 on the NTD and connects it to TM1 (residues 263 to 287). The flexing of the NTD towards the membrane involves significant conformational rearrangement of the proline-rich linker, which is capped by residues 243. We propose that the appearance of the final allosteric node in the network: fragment *f242*, is asso-ciated with this rearrangement (S5 Fig). Very recently, Saha et al. [79] introduced a disulfide bridge between the linker (residue Pro251) to the CTD (residue Leu929) to constrain the dynamics of the NTD, and this did not interfere with the cholesterol transport activity of NPC1. However, bending of the NTD to the extent observed in our simulations can still be driven by the flexibility of residues upstream of Pro251 in the linker (S5 Fig).

In summary, the interactions between the luminal CTD, MLD and NTD domains are rather dynamic in the absence of cholesterol in the SSD, and this enables the NTD to intermittently disconnect from the rest of the protein and thereby come closer to the membrane. In the pres-ence of an SSD-bound cholesterol, however, this interaction network is stabilised, and prevents the free movement of the NTD.

## An NTD-bound cholesterol suppresses the allosteric network

So far, we investigated the dynamics of NPC1 before the putative transfer of a cholesterol mol-ecule from NPC2 to the NTD. To investigate the change in conformational dynamics upon loading the NTD with a cholesterol molecule, we launched an additional set of simulations with an NTD-bound cholesterol, while keeping all the other simulation parameters identical. Five replicates (200 ns each) for each setup (termed POPC-x, POPC-CHOL-x, and POPC-CHOL-bound-x) were simulated. The differences between the dynamics of the NTD in the POPC-x, POPC-CHOL-x and POPC-CHOL-bound-x simulation are significantly suppressed compared to those observed with a cholesterol-free NTD reported earlier. The NTD no longer tilts for the POPC-x simulations, and the inter-domain interaction networks described in the previous section are altered (Fig 10). For example, the NTD makes two strong salt bridges with the CTD in the POPC-x simulations (S12 Fig, which were not observed for the POPC simula-tions. In addition, the 2D-scatter plot of conformations projected upon the first two eigenvec-tors of the POPC-x simulation does not show differences (S11 Fig) as significant as those

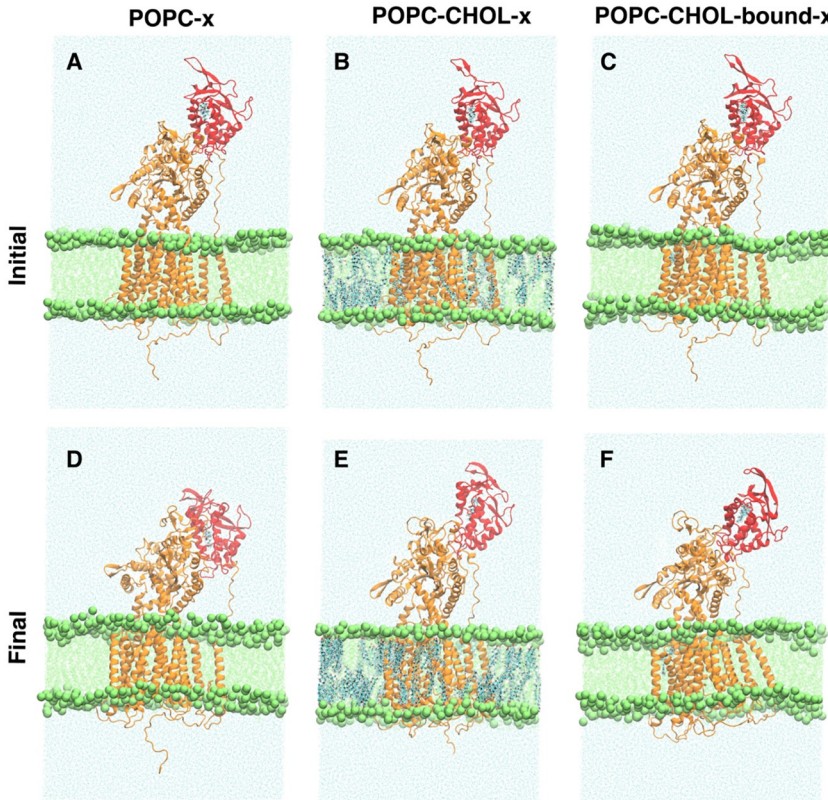

**Fig 10. Representative initial and final simulation snapshots for one of the POPC-x (A and D), POPC-CHOL-x (B and E), and POPC-CHOL-bound-x (C and F) simulations, with an NTD-bound cholesterol.** The NTD is depicted in red.

presented in (Fig 4). The implications of the altered dynamics of the NTD in the presence of an NTD-bound cholesterol are discussed further in the next section.

## Discussion

NPC1 is a putative sterol transporter in the lysosome with at least two binding sites for cholesterol, one in the NTD, and another in the SSD. While cholesterol binding to the NTD of NPC1 has been implicated in the cholesterol transport function of the protein, the role played by cholesterol binding to the membrane embedded SSD has remained obscure. Here, we show that the binding of a single cholesterol molecule to the SSD in NPC1 has a remarkable allosteric consequence for the conformations of the luminal domains, in particular of the NTD (Fig 11). In the presence of cholesterol in the SSD, a salt bridge between Asp620 and Lys1217 stabilises the upright conformation of the NTD by the network of interactions between the luminal domains as described above. In the absence of cholesterol, the salt bridge breaks, resulting in the unfolding of TM3, leading to the severing of MLD-CTD contacts, in turn severing contacts between the CTD and the NTD. In this way, the NTD is released from the rest of the luminal domains, and approaches the membrane in some of the POPC simulations. The binding of cholesterol in the SSD and the consequent stabilisation of the Asp620-Lys1217 salt bridge has a profound impact on NTD conformational dynamics.

Allostery has long been considered to take place as a series of discrete changes in a protein conformation upon binding of an effector molecule, often at a distal site in a protein. Recent

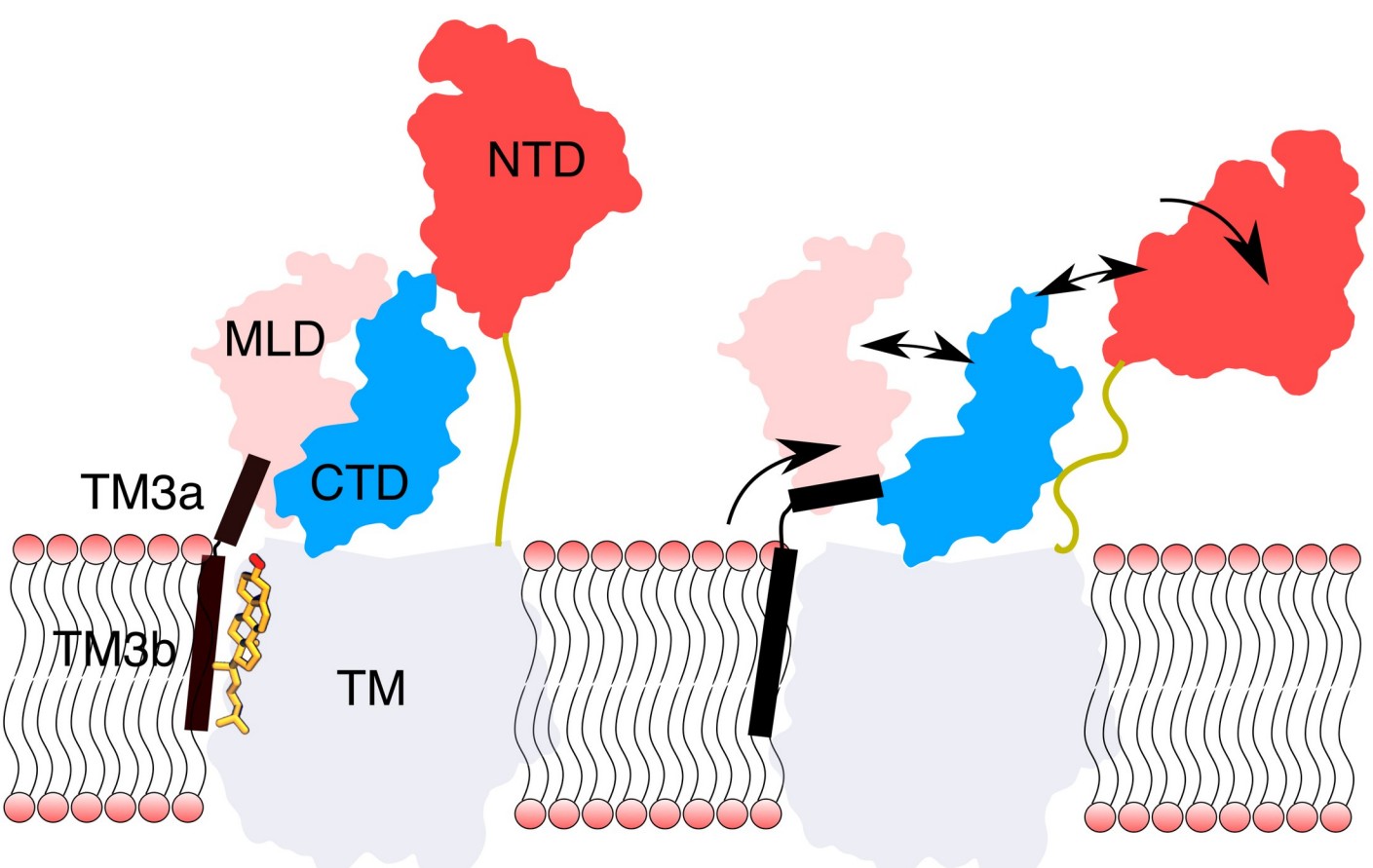

**Fig 11. Proposed allosteric mechanism.** In the presence of an SSD-bound cholesterol (left), TM3a and TM3b form a nearly continuous helix. Without a cholesterol in the SSD, a salt bridge between Arg620 and Lys1217 breaks, (Fig 7B) and TM3 unfolds near Asp620. The unfolding of TM3 triggers a series of events which breaks contacts between the luminal domains, ultimately disengaging the NTD from the rest of the protein.

studies, however, emphasize the ensemble view and the role played by altering the conformational dynamics of a protein upon ligand binding in driving allosteric transitions (e.g. [80]). According to the ensemble view, allostery can take place entirely without gross conformational changes when the dynamics of a protein gets altered. Such transmission of allosteric signals can occur in some systems, for example, by the organisation of semi-rigid domains or subunits interacting via flexible regions, such that local binding events propagate over a long distance ([81]), as noted by [82]. In fact, the ensemble view was proposed 35 years ago in a theoretical model put forward by Cooper and Dryden [83] who suggested that minute but global "rigidification" of a protein by a ligand can cause positive co-operativity and allosteric transitions. The results as we present here fit very well in the ensemble view. We demonstrate how cholesterol acts an effector by reducing the conformational dynamics of NPC1. The binding of cholesterol shifts the ensemble of possible conformations by stiffening a few selected modes, which we elucidate in detail. The allosteric signal passes through the semi-rigid CTD, MLD and NTD domains, via flexible regions which we characterize as key nodes in the allosteric path. We provide a pathway for signal transmission from the SSD to the NTD and suggest that this process is instrumental for cholesterol pick up in the NTD at the luminal side of the lysosome.

What is the mechanistic relevance of allosteric confinement of the NTD in NPC1 upon binding of cholesterol in its transmembrane sensing domains? A definite answer to this

question requires knowledge of the exact mechanism with which NPC1 picks up cholesterol from NPC2 and shuttles it into the lysosomal membrane. Despite significant progress in the description of the molecular details of NPC1-driven cholesterol transport in the last few years [24], the mechanism of cholesterol pickup and transfer is not known definitively. We can envision a dual role for the allosteric network we discovered in NPC1 in the process of cholesterol insertion into the lysosomal membrane: on one hand, binding of cholesterol to the SSD suppresses cholesterol transfer from NPC2 to the NTD (negative feedback onto the pre-loading state). On the other hand, SSD-bound cholesterol reinforces delivery of cholesterol from the NTD into the lipid bilayer (positive feedback on the post-loading state). If no cholesterol is bound to either the NTD or the SSD, the NTD is free to bend and thereby to open the entrance of the sterol binding pocket for pick up of a cholesterol molecule delivered by NPC2. Once the NTD has received a cholesterol molecule, it orients upright likely allowing for sterol transfer through the protein and insertion into the lysosomal membrane. Support for this assertion is obtained from our observed enrichment of the upright conformation in the POPC-x simulations (Fig 10). We do not observe the same enrichment in the POPC-CHOL-x or POPC-CHOL-bound-x simulations, suggesting that further investigations are required to fully resolve the allosteric effects of an NTD-bound cholesterol in concert with an SSD-bound cholesterol.

During the preparation of this manuscript, several new developments support the existence of a tunnel passing through the luminal domains in NPC1, along which a cholesterol molecule can slide all the way from the NTD to the SSD. For the NPC systems, the hypothesis was first made for the *Saccharomyces cerevisiae* NPC protein NCR1, where the NTD is positioned to deliver a cholesterol molecule to the luminal domains [84]. Later, a cryo-EM structure of NPC1 [40] showed the inhibitor itraconazole blocking the putative tunnel leading to the SSD, lending further support to the tunnel transfer mechanism. Simulations of a wild-type variant of NPC1 containing the mutation L472P postulated that the cholesterol transport functionality of the mutant is compromised because the tunnel is disrupted [85]. Similarly, constraining the dynamics of the NTD does not interfere with cholesterol transport activity, while locking the MLD and CTD domains does [79], lending further support to the transfer through a tunnel. It is worth noting that modifying inter-domain interactions can modulate not only cholesterol transfer through a tunnel, but also by any other mechanism involving concerted domain interactions, such as a transfer of cholesterol to a different NPC1 molecule, or even a direct transfer to the membrane [86]. Limited movement of cholesterol along the putative tunnel was observed in a 100 ns simulation of NPC1 carrying the P691S mutation [87]. However, longer simulations with multiple replicas are required to confirm the statistical relevance of this observation.

Transfer of cholesterol through the tunnel necessitates that the NTD remains upright to hand cholesterol over to the other luminal domains. Upon release of sterol from the NTD but prior to its binding to the SSD, the NTD can disengage again and pick up a new cholesterol molecule from NPC2. Transport of the latter into the membrane is reinforced by binding of the first cholesterol molecule to the SSD, as it keeps the NTD in the upright position with its sterol binding pocket oriented towards the protein. Once a cholesterol molecule dissociates from the SSD, the emptied NTD can flex again to allow for transfer of a new cholesterol molecule from NPC2. Support for such a mechanism is provided by our observation that the NTD tilts when cholesterol leaves the SSD binding site in one of the POPC-CHOL-bound simulations. (S1 Fig). Furthermore, in the flexed conformation, the cholesterol binding pocket of the NTD is exposed to the lumen, and is primed to receive a cholesterol molecule from NPC2 (S5 Fig).

Note that NPC2 can also deliver cholesterol to other proteins such as the glycoprotein LAMP2 in the lysosomal membrane [88]. In this scenario, membrane cholesterol would lower

the likelihood of pick up (which requires NTD flexibility) and increase the likelihood for sterol membrane insertion (which needs NTD confinement). Thus, the kinetics of sterol binding to and dissociation from the SSD can play a key role in the control of the allosteric conformational change in NPC1 during the cholesterol transport cycle.

The cholesterol-binding to the SSD is a second order reaction, hence the rate constant for binding ($k_{on}$) is bimolecular, i.e., ligand concentration dependent. Accordingly, increasing the abundance of the ligand cholesterol in the membrane will speed up the binding process (increase $k_{on}$). The release rate constant, $k_{off}$ is however monomolecular. Accordingly, increasing cholesterol in the membrane will greatly influence the binding rate without affecting cholesterol release with the net effect that cholesterol will always be SSD-bound in a cholesterol enriched lysosomal membrane. In this way, the cell can fine tune the binding affinity of cholesterol to the SSD of NPC1 by regulating cholesterol abundance in the lysosomal membrane. Several proteins have been suggested to receive cholesterol from NPC1 for export to other cellular membranes, like the ER or the plasma membrane [89]. We speculate that such protein-mediated outflow of cholesterol from the lysosomal membrane contributes to a relatively constant cholesterol level and to setting the allosteric threshold for NPC1 function.

In conclusion, we demonstrate that without cholesterol, the luminal domains of NPC1 are highly dynamic and enable to the NTD to disengage from the rest of the protein, and bend towards the membrane. An SSD-bound cholesterol operates as a **conformational brake** which restricts the dynamics of the NTD. We propose that cholesterol acts as an allosteric effector upon binding to the SSD, and the response of the luminal domains to cholesterol binding in the membrane constitutes a feedback mechanism which maintains cholesterol homeostasis in the lysosome. Mechanistically, we propose that the flexed conformation of the NTD is primed to receive a cholesterol from NPC2. In the upright conformation, the NTD is likely to facilitate the transfer of the cholesterol molecule through the tunnel.

The simulations were performed with NPC1 embedded in a simple POPC bilayer to model the lysosomal membrane, which has a much more complex composition also containing sphingolipids, which can bind cholesterol, and potentially influence NPC1-membrane interactions and modulate the conformational dynamics of the protein. Additionally, the lysosomal membrane is asymmetric, containing different types of lipids in the two leaflets. There is also a pH gradient across the lysosomal membrane. In our simulations, periodic boundary conditions impose the constraint that the luminal and cytosolic compartments are identical, although the initial protonation states of the luminal amino acid residues are chosen based on a pH of 5.0. The glycocalyx which can possibly contain charged sugars such as sialic acid, is also not modelled here. All of these complexities can alter the conformational dynamics of NPC1. Information about residence times and binding kinetics of cholesterol in the SSD can also be gleaned from Markov state Models which necessitate a much larger number of simulations of longer length ([90]). Recent strong biochemical experiments and structural evidence for the presence of sterols or inhibitors in the tunnel cavity supports the existence of tunnels through NPC1, NCR1 and Patched. The software Mole [91] is often used to visualise the tunnels. During our independent calculations of tunnels from these structures, we found that Mole often reported several tunnels of different lengths and radii passing through various cavities of the protein. Furthermore, we were often unable to obtain a single continuous tunnel pathway through the protein without introducing user bias in the software, such as the merging of different tunnels or optimising input parameters to visualise a specific tunnel. Pending resolution of these issues, we have not included the tunnel data in the current manuscript, and propose that future simulation efforts to investigate the tunnel could focus on calculating the free energy of cholesterol transfer using, for example, Potential of Mean Force calculations.

## Supporting information

**S1 Fig.** (A) The distance between cholesterol and the centre of mass of the cholesterol-binding residues in the binding site as shown in Fig 1, for the 10 different simulation replicas of the POPC-CHOL-bound simulations. Cholesterol temporarily escapes the binding pocket in replicas 3,4,5 and 8. However, in replicas 3 and 4, cholesterol returns to the binding site within 200 ns. (B) Simulation snapshot from sim4, showing that the NTD tilts when cholesterol leaves the SSD binding site.
(TIF)

**S2 Fig.** (A), (B) and (C) Root mean squared deviation (RMSD) from the initial structure for all simulations. S1 through S10 denote the 10 different simulation replicates. (D) Histogram of the RMSD.
(TIF)

**S3 Fig. Statistical assessment of the angle distribution for the angle $\alpha$ in Fig 4E.** Two approaches were used: manual fitting and fitting using a Gaussian mixture model. (A) Manual fitting using the sum of 3 Gaussians. Initial conditions for the fitting were chosen by inspecting the distribution. The function and fit were refined by using the Akaike Information Criterion/ Bayesian Information Criterion (BIC). (B) Means and standard deviations of each Gaussian obtained from manual fitting. (C) Clusters obtained from the Gaussian Mixture model mapped onto the angle $\alpha$ time series. (D) Means and standard deviations of the clusters. (E) Comparison of the results for cluster 2 and Gaussian 3 (i.e. representatives of the flexed state) obtained from manual fitting and Gaussian Mixture model.
(TIF)

**S4 Fig. Multiple sequence alignment for NPC1 in the SSD near Glu688.** The different colours in the alignment correspond to conservation of different types of residues.
(TIF)

**S5 Fig. TM1 is connected to residues 242 to 245 on fragment *f242* by a flexible proline-rich spacer (shown in yellow), the conformation of which changes significantly in the POPC simulation (A), but less so when the NTD does not tilt in the POPC-CHOL (B) and POPC-CHOL-bound (C) simulations.** The black arrow shows the putative entry point of a sterol in the NTD [18].
(TIF)

**S6 Fig. The average distance matrix for the entire protein for the POPC systems.** The matrix is obtained by calculating average distances between residue pairs. Only residue pairs within a cut-off distance of 0.5 *nm* are shown.
(TIF)

**S7 Fig. The average distance difference matrix for the entire protein.** The matrix is obtained by subtracting average distances of the POPC-CHOL simulations from the average distances of the POPC simulations.
(TIF)

**S8 Fig.** (A) Percentage helical content for TM3 compared for all three sets of simulations. (B) Distribution of the angle between TM3a (residues Asp620 to Leu639 and TM3b (residues Thr604 to Arg615) for all 10 POPC, POPC-CHOL and POPC-CHOL-bound simulations. The difference between the distributions is not as apparent here as it is in Fig 7.
(TIF)

**S9 Fig. Tyr634 on TM5 interacts with Phe1207 and Leu1204 on TM12, but the interaction is weaker for the POPC system.** (A) Radial distribution functions between Tyr634 and Leu1204. (B) Radial distribution functions between Tyr634 and Phe1207 (C) and (D) Simulation snapshots from the POPC and POPC-CHOL systems respectively.
(TIF)

**S10 Fig. Radial distribution function between the center of mass of the NTD and the CTD.** The two domains move apart in the POPC simulations on an average, owing to the flexing of the NTD towards the membrane.
(TIF)

**S11 Fig. Principal component analysis for the simulations with an NTD-bound cholesterol.** Scatter plot of the all trajectory frames projected on first two eigenvectors of the POPC-x simulation. The analysis is performed on concatenated trajectories of 5 independent 200 ns simulations in each case.
(TIF)

**S12 Fig.** The salt-bridges between the NTD and the CTD in the POPC-x simulations: (A) Radial distribution function between the center of masses of Asp85 and Lys1010. The salt bridge only exists in the POPC-x simulations (B) Radial distribution function between the center of masses of Glu61 and Arg978. The salt bridge is much stronger for the POPC-x simulations (C) and (D) Initial and final POPC-x simulation snapshots corresponding to the interactions in A. (E) and (F) Initial and final POPC-x simulation snapshots corresponding to the interactions in B. The analysis is performed on concatenated trajectories of 5 independent 200 ns simulations.
(TIF)

**S1 Video. Sample simulation trajectories from the three different types of simulations.** The NTD bends significantly in the POPC simulations.
(MP4)

**S2 Video. Sample trajectories projected upon the first two eigen vectors from the PCA analysis of the POPC simulations.** The NTD bends significantly in the POPC cases.
(MP4)

## Author Contributions

**Conceptualization:** Daniel Wüstner, Himanshu Khandelia.

**Data curation:** Vikas Dubey, Behruz Bozorg, Himanshu Khandelia.

**Formal analysis:** Vikas Dubey, Behruz Bozorg.

**Funding acquisition:** Himanshu Khandelia.

**Investigation:** Vikas Dubey, Himanshu Khandelia.

**Methodology:** Vikas Dubey, Behruz Bozorg, Daniel Wüstner, Himanshu Khandelia.

**Project administration:** Himanshu Khandelia.

**Resources:** Himanshu Khandelia.

**Supervision:** Daniel Wüstner, Himanshu Khandelia.

**Visualization:** Vikas Dubey.

**Writing – original draft:** Vikas Dubey, Behruz Bozorg, Daniel Wüstner, Himanshu Khandelia.

**Writing – review & editing:** Vikas Dubey, Daniel Wüstner, Himanshu Khandelia.

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
