## [Decision Letter · Decision Letter 0]

5 Feb 2020

Dear Dr Khandelia,

Thank you very much for submitting your manuscript "Cholesterol binding to the sterol-sensing region of Niemann Pick C1 protein confines dynamics of its N-terminal domain" for consideration at PLOS Computational Biology.

As with all papers reviewed by the journal, your manuscript was reviewed by members of the editorial board and by several independent reviewers. In light of the reviews (below this email), we would like to invite the resubmission of a significantly-revised version that takes into account the reviewers' comments.

We cannot make any decision about publication until we have seen the revised manuscript and your response to the reviewers' comments. Your revised manuscript is also likely to be sent to reviewers for further evaluation.

Sincerely,

Bert L. de Groot

Associate Editor

PLOS Computational Biology

Nir Ben-Tal

Deputy Editor

PLOS Computational Biology

Reviewer's Responses to Questions

**Comments to the Authors:**

Reviewer #1: Dubey et al. describe an extensive atomistic MD simulation study of

the Niemann Pick C1 (NPC1) membrane protein in a pure POPC bilayer, in

POPC-Cholesterol mixture, and in pure POPC with a cholesterol molecule

bound to the sterol sensing domain (SSD) within the membrane (each 10

simulations of 200ns). In three of 10 simulation without cholesterol,

a slight tilting of the luminal N-terminal domain towards the membrane

was observed. This observation of a cholesterol-dependent degree of

conformational flexibility of the NTD is hypothesized to display an

allosteric effect of cholesterol.

Overall, the simulations are state-of-the-art. However, the

statistical significance of the observation remains unclear. Also, it

is unclear how the small additional tilting of the NTD in absence of

cholesterol contributes to the function of NPC1; the NTD is still far

away from the membrane surface rendering a direct transfer of

cholesterol from the NBD to the membrane unlikely. Additionally, an

NTD-bound cholesterol was not even considered in the

simulations. Similarly, a second discussed option, a possibly

NTD-assisted transport of cholesterol in the initial upright position

was not investigated in this study.

Main points:

The simulation structure is based on a composed model of a crystal

structure, an additional modeled transmembrane helix and NTD (from

cryo-EM). The stability of the structure during the equilibration was

hardly addressed in the manuscript. However, also the

cholesterol-bound simulations show substantial deviations from the

initial structure (rmsd plots in Fig. S1).

In three out of ten simulations an increased tilting of the NBD

towards the membrane was observed. For which percentage of overall

simulation time was this observed? Fig.4A suggests that this was the

case for a very limited time, only. Also, after tilting towards the

membrane, did the NTD swivel back during simulation?

p5: The authors claim that "the NTD distance for POPC-CHOL appears to

be about 1 nm greater compared to POPC in the distance distribution

(Figure 3A)" and that "the distance between NTD and membrane is

significantly higher" for the POPC-CHOL-bound simulation as compared

to POPC-CHOL. These findings can hardly be seen in Fig. 3. With

membrane cholesterol the distance distribution may be narrowed, but a

shift is not visible.

It would be great if the authors would include a scheme displaying the

conformational transitions along the computed eigenvectors.

p11, statistics: "Many of the amino acid residues belonging to the

fragments implicated in the allosteric 289 path are conserved across

Eukarya." How many of them are conserved (%)? If randomly choosing

fragments, how large is the conservation then?

The authors discuss in detail the impact of bound cholesterol on

residue-residue interactions. This is done by comparing distances from

three (out of ten) simulations with tilted NBD domain (without

cholesterol to all simulations with cholesterol. Many interactions are

declared to be weakened and discussed. However, an overall picture is

lacking how binding of cholesterol restricts the conformational space

of NBD. I.e., what is the sequence of events? Which one is the

decisive step, why is a small tilting only observed in three of the

ten simulations?

Minor points:

p5, line 170: Fig. 3A should be Fig. 3D

Figure 3 is not fully displayed in manuscript

Fig. 4B,D different lines should be explained in the caption

p10-15. The results section suffers from the repeated discussion of

numbered fragments that are difficult to relate to structure; It is

probably easier to directly name segments by residue numbers.

Reviewer #2: In this work Dubey et al. study how Cholesterol binding to the sterol-sensing region of a Niemann Pick C1 protein confines the motion its N-terminal domain. The work's take home message is rather striking and inspiring: Binding of a single cholesterol molecule to a distal cluster of transmembrane domains can trigger a whole allosteric 'chain reaction' that restricts large conformational changes of a distal n-terminal domain. Furthermore, the resolved allosteric mechanism/pathway is extensively compared to known putative point mutations and conserved residues. I believe these insights to be highly valuable for the field of structural biology since it can causally explain the relationships between them. Finally, the performed simulations and analysis are both solid and state-of-the-art. I have only few minor comments on this work:

Page 5 in the title: .....cholesterol-dependant -> dependent

Page 8: "and the rate constant for binding is bimolecular."

I do not follow the flow of logic or implication of this statement. Yes. There are only two molecular

species, but why one rate constant? Is the binding competitive? Are lipids able to bind into that pocket as well, however, with a far weaker

afinity. Hence, cholesterol is not knocked out from the binding pocket in the cholesterol free simulation.

Page 8: "We simulated an extra system with a different cholesterol distribution in

the membrane, and in this case too, cholesterol bound to the SSD on a timescale of"

Distribution -> Composition? Note that this extra system is also nowhere described.

Page 8: Does cholesterol ever unbind? Worth looking at how long cholesterol stays bound in the POPC-CHOL-bound system?

Perhaps add an explicit statement about this since the reader will be wondered about this.

Page 12: Fig.4. Description of figure E is missing within the caption.

Inconsistent use of capital letters throughout the manuscript. For example:

Page 8: "Covariance and principal components Analysis" Check capital letters.

Page 9: "between two vectors: one vector" -> :One vector (capital)

Page 10: Fig 2 -> "Representative Initial" -> .....initial

Page 11: origin in Figure 4A and Figure 4C. Check whether the F in 'Figure' must capitalized when not being an abbreviation. One page later, on page 15, l296: "Supplementary figure" ...figure is written with a small "f"

**Have all data underlying the figures and results presented in the manuscript been provided?**

Reviewer #1: Yes

Reviewer #2: Yes

PLOS authors have the option to publish the peer review history of their article (what does this mean?). If published, this will include your full peer review and any attached files.

Reviewer #1: No

Reviewer #2: No
---

## [Decision Letter · Decision Letter 1]

2 Jul 2020

Dear Dr Khandelia,

We are pleased to inform you that your manuscript 'Cholesterol binding to the sterol-sensing region of Niemann Pick C1 protein confines dynamics of its N-terminal domain' has been provisionally accepted for publication in PLOS Computational Biology.

Best regards,

Bert L. de Groot

Associate Editor

PLOS Computational Biology

Nir Ben-Tal

Deputy Editor

PLOS Computational Biology

Reviewer's Responses to Questions

**Comments to the Authors:**

Reviewer #1: The Authors did a good job for the revision and tackled almost all raised points.

**Have all data underlying the figures and results presented in the manuscript been provided?**

Reviewer #1: None

PLOS authors have the option to publish the peer review history of their article (what does this mean?). If published, this will include your full peer review and any attached files.

Reviewer #1: No

---

## [Editor Report · Acceptance letter]

11 Sep 2020

PCOMPBIOL-D-19-01892R1

Cholesterol binding to the sterol-sensing region of Niemann Pick C1 protein confines dynamics of its N-terminal domain

Dear Dr Khandelia,

I am pleased to inform you that your manuscript has been formally accepted for publication in PLOS Computational Biology. Your manuscript is now with our production department and you will be notified of the publication date in due course.

With kind regards,

Matt Lyles
